# Genomic Surveillance of SARS-CoV-2 in Healthcare Workers: A Critical Sentinel Group for Monitoring the SARS-CoV-2 Variant Shift

**DOI:** 10.3390/v15040984

**Published:** 2023-04-17

**Authors:** Dayane Azevedo Padilha, Doris Sobral Marques Souza, Eric Kazuo Kawagoe, Vilmar Benetti Filho, Ariane Nicaretta Amorim, Fernando Hartmann Barazzetti, Marcos André Schörner, Sandra Bianchini Fernandes, Bruna Kellet Coelho, Darcita Buerger Rovaris, Marlei Pickler Debiase Dos Anjos, Juliana Righetto Moser, Fernanda Rosene Melo, Bianca Bittencourt De Souza, Dimitri da Costa Bessa, Fernando Henrique de Paula e Silva Mendes, Alexandra Crispim Boing, Antonio Fernando Boing, Josimari Telino de Lacerda, Guilherme Valle Moura, Daniela Carolina De Bastiani, Milene Höehr de Moraes, Luiz Felipe Valter De Oliveira, Renato Simões Moreira, Patricia Hermes Stoco, Maria Luiza Bazzo, Gislaine Fongaro, Glauber Wagner

**Affiliations:** 1Centro de Ciências Biológicas, Universidade Federal de Santa Catarina, Florianópolis 88040-900, Santa Catarina, Brazil; dayufsc@gmail.com (D.A.P.); doris.sobral@gmail.com (D.S.M.S.); kazuo.eric@gmail.com (E.K.K.); vilmarbf98@gmail.com (V.B.F.); ariane.nicarettaamorim@gmail.com (A.N.A.); patricia.stoco@ufsc.br (P.H.S.); gislaine.fongaro@ufsc.br (G.F.); 2Centro de Ciências da Saúde, Universidade Federal de Santa Catarina, Florianópolis 88040-370, Santa Catarina, Brazil; fernandohb55@hotmail.com (F.H.B.); marcos.schorner@gmail.com (M.A.S.); alexandra.boing@ufsc.br (A.C.B.); antonio.boing@ufsc.br (A.F.B.); josimari.telino@ufsc.br (J.T.d.L.); marialuizabazzo@gmail.com (M.L.B.); 3Laboratório Central do Estado da Saúde de Santa Catarina, Florianópolis 88010-001, Santa Catarina, Brazil; genomicalacensc@gmail.com (S.B.F.); brunakellet@gmail.com (B.K.C.); darcitarovaris@gmail.com (D.B.R.); marleipickler@saude.sc.gov.br (M.P.D.D.A.); 4Diretoria de Vigilância Epidemiológica de Santa Catarina, Florianópolis 88015-130, Santa Catarina, Brazil; variantescovid.sc@gmail.com (J.R.M.); fernandar.melo@gmail.com (F.R.M.); bittenka@gmail.com (B.B.D.S.); dimitribessa@gmail.com (D.d.C.B.); fernandohpsm@hotmail.com (F.H.d.P.e.S.M.); 5Centro de Sócio Econômico, Universidade Federal de Santa Catarina, Florianópolis 88040-900, Santa Catarina, Brazil; guilherme.moura@ufsc.br; 6Biome-Hub Pesquisa e Desenvolvimento, Florianópolis 88054-700, Santa Catarina, Brazil; daniela.bastiani@biome-hub.com (D.C.D.B.); milene.hoehr@biome-hub.com (M.H.d.M.); felipe@biome-hub.com (L.F.V.D.O.); 7Instituto Federal de Santa Catarina, Gaspar 89111-009, Santa Catarina, Brazil; renatosm@gmail.com

**Keywords:** healthcare workers, SARS-CoV-2 variants, Omicron, sentinel groups, COVID-19

## Abstract

SARS-CoV-2 genome surveillance is important for monitoring risk groups and health workers as well as data on new cases and mortality rate due to COVID-19. We characterized the circulation of SARS-CoV-2 variants from May 2021 to April 2022 in the state of Santa Catarina, southern Brazil, and evaluated the similarity between variants present in the population and healthcare workers (HCW). A total of 5291 sequenced genomes demonstrated the circulation of 55 strains and four variants of concern (Alpha, Delta, Gamma and Omicron—sublineages BA.1 and BA.2). The number of cases was relatively low in May 2021, but the number of deaths was higher with the Gamma variant. There was a significant increase in both numbers between December 2021 and February 2022, peaking in mid-January 2022, when the Omicron variant dominated. After May 2021, two distinct variant groups (Delta and Omicron) were observed, equally distributed among the five Santa Catarina mesoregions. Moreover, from November 2021 to February 2022, similar variant profiles between HCW and the general population were observed, and a quicker shift from Delta to Omicron in HCW than in the general population. This demonstrates the importance of HCW as a sentinel group for monitoring disease trends in the general population.

## 1. Introduction

According to the World Health Organization (WHO), as of March 2023 (37 months after the beginning of the pandemic caused by SARS-CoV-2), COVID-19 was responsible for approximately 759 million confirmed cases and around 6.8 million deaths worldwide. In Brazil, there have been approximately 37.1 million confirmed cases of COVID-19 and more than 699,000 deaths, representing 9.2% of the global death toll [1].

Healthcare workers (HCW) have been the frontline response to the COVID-19 pandemic and have been significantly affected by the disease, mainly during the first epidemic wave [2]. A meta-analysis of prevalence conducted across 97 studies involving HCW infected with SARS-CoV-2 showed that nurses, healthcare assistants, and physicians were the most affected by working in non-emergency wards, in screening, and with hospitalized patients [2]. During the first wave in Western Sicily and Italy, HCW were more susceptible to infection than the general population, with over 95% working in non-COVID-19 environments [3]. A study conducted in the Puglia region of Italy compared the impact of three waves of COVID-19 on HCW and found that the second wave had a significant effect. This was likely due to the less stringent lockdown measures during that period, which allowed for greater transmission of SARS-CoV-2. However, they observed that the positivity rate among doctors decreased from the first to the third wave (from 59 to 24%), which was attributed to increased adherence to protection measures, such as Personal Protective Equipment (PPEs) and the availability of vaccines [4]. In addition, a seroprevalence study performed among 4987 HCW in São Paulo, Brazil, during the first wave of the COVID-19 pandemic showed a 14% serological positivity for SARS-CoV-2 (symptomatic/asymptomatic). This was higher than the rates reported for HCW in Germany (10.7%) and China (0.2%) [5,6,7].

Despite the lack of SARS-CoV-2 sequencing in HCW, this group represents a significant risk factor for SARS-CoV-2 infection. The HCW are constantly exposed to SARS-CoV-2, making them more susceptible to infection from the various viral variants circulating in the general population. However, they also have greater access to official testing services for monitoring, which helps avoid the underreporting of cases. It is important to notice that certain characteristics of HCW reinforce the importance of establishing sentinel groups. These characteristics include agility in monitoring, population representation, possibility of data extrapolation, and access to real-time data. This will enable better decision making in the fight against infectious diseases [8,9]. Therefore, HCW represent a good sentinel group for genomic SARS-CoV-2 surveillance.

A study published by Bromberg et al. (2022) demonstrated the importance of controlling the COVID-19 disease in Israel by monitoring the pandemic and consequently reducing the population’s search for medical care [10]. SARS-CoV-2 genomic surveillance has shown a relevant impact since the beginning of the COVID-19 pandemic in 2020, when it was first used to monitor the spread of the pathogen. By February 2020, the first study describing the first likely patient hospitalized due to an unknown virus, later known as SARS-CoV-2, had been published [11]. Ever since the first genome sequencing, genome surveillance has been used to understand the evolution, genetic diversity, and epidemiology (including transmissibility, mortality, and prognosis) of SARS-CoV-2 [12]. By March 2023, approximately 13.7 million SARS-CoV-2 genomic sequences were available in the GISAID database. The main objective of this monitoring was to track genomic mutations and the dispersion of SARS-CoV-2 variants [13].

During the COVID-19 pandemic, mutations in the SARS-CoV-2 genome have contributed to changes in pandemic dynamics caused by the emergence of variants that have provided new pandemic waves. Garcia-Beltran et al. (2022) compared the infection rates of the main SARS-CoV-2 variants to the wild type and observed that there were few differences between the Gamma and Beta variants compared to the wild type virus rates [14]. However, the Delta and Omicron variants showed significantly higher infection rates, almost double and quadruple the wild type rate, respectively. Another SARS-CoV-2 monitoring study demonstrated its epidemiological importance by identifying a P.1-Like-II subvariant isolated in western Santa Catarina state, correlating it with an increase in the number of deaths in the same region and period [15].

Therefore, this study aimed to evaluate the dispersion of SARS-CoV-2 among the general population in Santa Catarina, Brazil, from May 2021 to April 2022 and compare it to that in HCW during the Omicron wave to understand the role of HCW as sentinels in human viral surveillance.

## 2. Materials and Methods

### 2.1. Studied Regions and Sampling

This study was conducted in the state of Santa Catarina (SC), southern Brazil, between the second and third years of the pandemic, from May 2021 to April 2022. During this period, there were 454,732 confirmed cases and 3875 deaths owing to COVID-19 in Santa Catarina. Out of 5181 sequences deposited in GISAID (dataset available under the EPI_SET ID, *EPI_SET_230227qf*), our research group sequenced 2080 randomly selected samples (October 2021 to April 2022). In addition, 110 random samples from HCW from 12 cities in Santa Catarina were sequenced between November 17, 2021 and February 21, 2022 (dataset available under the EPI_SET ID, *EPI_SET_230227vt*). On a weekly basis, samples were randomly selected from the population of individuals who tested positive for SARS-CoV-2 using RT-qPCR, and weekly sample sizes were defined to estimate the proportions of SARS-CoV-2 variants in the population with a 10% margin of error and 95% confidence.

The study was conducted in accordance with the Declaration of Helsinki and approved by the Ethics Committee on Research Humans Beings of the Federal University of Santa Catarina (Assent number: 5.071.435, CAAE:52676721.8.0000.0121, 29 October 2021). Only public data available between May 2021 and September 2021 on the GISAID platform were analyzed.

### 2.2. SARS-CoV-2 Genome Sequencing

A total of 2190 nasopharyngeal swab samples, EPI_SET_230321ms, were obtained from SARS-CoV-2 positive patients and HCW diagnosed using RT-qPCR. The inclusion criterion was a positive RT-qPCR for gene E, with Ct values of <25. Sampling took place between May 2021 and April 2022 for the general population (2080 samples), and between November 2021 and February 2022 for HCW (110 samples). Total viral RNA was extracted using the RNA Viral Kit (QIAGEN, Hilden, Germany), and the Illumina COVIDSeq was used [16]. The libraries were prepared using the Illumina COVIDSeq protocol (Illumina Inc, San Diego, CA, USA). The primer pool also contained primers targeting human RNA, producing 11 amplicons. The PCR-amplified products were later processed for tagmentation and adapter ligation using IDT for Illumina Nextera UD Indexes Set A, B, C, and D (384 indexes, 384 samples).

Further enrichment and cleanup were performed according to the manufacturer’s protocols (Illumina Inc, San Diego, CA, USA). All samples were processed as batches in a 96-well plate that consisted of one COVIDSeq positive control HT (CPC HT) and one no-template control (NTC), and these 96 libraries were pooled together in a tube. Pooled samples were quantified using a Qubit 2.0 fluorometer (Invitrogen Inc., Waltham, MA, USA) and further using Collibri Library Quantification Kit for Illumina libraries (Thermo Fisher, Waltham, MA, USA). Fragment sizes were analyzed using an Agilent Fragment Analyzer 5200 (Agilent Inc., Santa Clara, CA, USA). The pooled library was further normalized to 650 pM concentration. For sequencing, pooled libraries were loaded onto the P1-flow cell following the NovaSeq-XP workflow, as per the manufacturer’s instructions (Illumina Inc, San Diego, CA, USA). Dual-indexed single-end sequencing with a 150 bp read length was performed on the NextSeq 1000 platform. Amplicon sequencing is a highly targeted approach that enables researchers to analyze genetic variations in specific genomic regions. Ultra-deep sequencing of PCR products (amplicons) allows for efficient identification and characterization of variants. This method uses oligonucleotide probes designed to target and capture regions of interest, followed by next-generation sequencing (NGS) [16].

### 2.3. Genome Sampling

A total of 6058 SARS-CoV-2 genome sequences from Santa Catarina (Brazil), collected between 1 May 2021 and 30 April 2022, with submission dates until 29 June 2022, were downloaded from GISAID. We selected 5181 sequences (*EPI_SET_230227qf*) based on genome coverage and location. Genomes with coverage below 90% and misplaced cities (erroneously labeled as Santa Catarina) were filtered out.

From these, 2080 (40.15%) were sequenced by our group according to the Illumina COVIDSeq protocol [16], originated and processed as previously described [15,16]. Samples from our group were labeled with the prefix *hCoV-19/Brazil/SC-UFSC*. Anonymized data on the number of COVID-19 cases and deaths were obtained from Boa Vista, an open data platform maintained by Santa Catarina (https://dados.sc.gov.br, accessed on 15 December 2022). In addition, 110 SARS-CoV-2 samples from HCW were processed as previously described [15,16]. These 110 samples (*EPI_SET_230227vt*) were analyzed in association with 698 genome sequences (*EPI_SET_230227ug*) generated by our research group that originated from the same cities as the HCW during the same period (November 2021 to February 2022).

The total number of HCW in Santa Catarina was obtained from public data available at the CNES database (https://elasticnes.saude.gov.br/profissionais, accessed on 15 December 2022). The samples were filtered using *nome_fantasia* (*hospital*, CEPON, LACEN, *secretaria de saúde* and *maternidade*), *profissional_atende_sus* (sim), *municipio*, and *profissional_cbo* (doctor, nurse, nurse assistant, and nursing technician). Redundancy was removed using HCW registration (*Professional-CNS*). An error rate of 9.31% was obtained at a 95% confidence level. The sample size was 110 and the total number of HCW (N) was 15,382, assuming a homogeneous distribution.

A comparison between HCW and the general population from November 2021 (epiweek 47 of 2021) to early February 2022 (epiweek 6 of 2022) was performed to determine the potential differences during the variant of concern (VOC) shift.

### 2.4. SARS-CoV-2 Variant Analysis

Amino acid mutations, deletions, and insertions were used to assess the dissimilarity of the genomes using the vegan R package v. 2.5.6 and the Jaccard index method. Multidimensional scaling (MDS) and the distribution of variants in Santa Catarina were plotted using the ggplot2 R package v. 4.2.0 [17] with samples present in *EPI_SET_230227vt* and *EPI_SET_230227ug*. Only one sample was assigned to the BA.2 lineage (*EPI_ISL_10077553*) and was removed from the Omicron MDS.

### 2.5. Maximum Likelihood (ML) Phylogenetic Analysis of HCW

Maximum likelihood (ML) analysis was performed for 808 samples (*EPI_SET_230227vt* and *EPI_SET_230227ug*) from HCW and the general population, in addition to the reference genome (GenBank accession number: NC_045512). Genome sequences were aligned by MAFFT v7.505 [18] using the FFT-NS-2 method, and an ML tree was constructed based on the multiple sequence alignment using IQ-TREE v2.2.0.3 [19]. The most suitable substitution model (GTR + F + R5) was predicted and automatically chosen using ModelFinder [20]. Ultrafast bootstrap [21] and SH-like approximate likelihood ratio tests (SH-aLRT) were performed with 1000 bootstrap replicates. The ML consensus tree was visualized and annotated using the iTOL web service [22].

## 3. Results

### 3.1. Profile of SARS-CoV-2 Variants in Santa Catarina (Brazil) during the Second Year of the COVID-19 Pandemic Period (May 2021 to April 2022)

The daily peaks of cases and deaths during the period of this study did not occur at the same time. From May 2021 to June 2021, the number of deaths and cases decreased (Figure 1A,B) as the shift in dominance from VOC Gamma to VOC Delta occurred (Figure 1C,D). However, between December 2021 and February 2022, the number of cases per day increased very quickly after the emergence of Omicron BA.1 by the end of November 2021, and the escalation of this variant was followed by the BA.2 sub-lineage in February 2022 (Figure 1C,D). Although the number of cases has significantly increased with the emergence of the Omicron variant around the beginning of 2022, daily deaths have only slightly increased and remain lower than those during the Gamma period (May 2021 to June 2021) (Figure 1A,B).

During this study, 55 different lineages and four VOCs—Alpha, Delta, Gamma, and Omicron (BA.1 and BA.2)—were identified (Figure 1C). The relative frequencies of the VOCs varied at different points during the study. In May 2021, the presence of the Gamma variant was higher, with the frequency decreasing until October 2021, whereas the presence of the Delta variant increased and persisted from July 2021 to January 2022. In December 2021, the Omicron variant became the dominant VOC in Santa Catarina (Figure 1D).

### 3.2. Variant Shift (Delta to Omicron) in HCW and General Population in Santa Catarina (Brazil)

At the end of the VOC Delta dominance and emergence of the VOC Omicron (November 2021 to February 2022), 26 lineages were identified and classified under both VOCs. Interestingly, the variant profiles were very similar when the relative frequencies in HCW and the general population were compared (Figure 2). However, it is worth noting that the shift among HCW occurred quicker than that in the general population, showing that the HCW group reproduced the pattern of the general population. In addition, multidimensional scaling (MDS) (Figure 3) and phylogenetic analyses (Appendix A) revealed that both the HCW and general population groups present similar SARS-CoV-2 genomes in all states of Santa Catarina (Figure 3A,B), reinforcing the idea that genomic surveillance in the HCW group also reflects the variants and genetic profiles circulating among the general population.

## 4. Discussion

Since 2020, SARS-CoV-2 has been responsible for numerous COVID-19 cases and related deaths worldwide. Even more than two years after the start of the pandemic, issues regarding rapid diagnosis, real-time monitoring, dissemination, and surveillance of the virus genome persist. Due to the higher cost of molecular techniques involved in SARS-CoV-2 diagnostics and sequencing, it is essential to find representative samples that can help reduce the costs associated with genomic surveillance. In September 2022, the coverage rate of SARS-CoV-2 was approximately 150 sequences per 100,000 inhabitants in the world and 60 sequences per 100,000 inhabitants in Brazil, demonstrating the low sequential coverage in Brazil relative to the world average [13,23].

This study aimed to monitor SARS-CoV-2 with the sequencing platform Illumina COVIDSeq (Illumina, San Diego, CA, USA) in Santa Catarina from May 2021 to April 2022, both in the general population and among HCW, evaluating HCW as possible sentinel groups.

In this study, we observed the presence of different variants in 12 months (May 2021 to April 2022) in Santa Catarina. Our group was responsible for sequencing 40.15% of the sequences submitted to GISAID for the same period and region, and it was also possible to evaluate the dispersion of the four VOCs (Alpha, Gamma, Delta, and Omicron—BA.1 and BA.2) (Figure 1).

In another study conducted by our research group during the first year of the pandemic, we identified the presence of two VOCs (Beta and Gamma) and 23 distinct lineages [15]. In the present study, 55 lineages and four VOCs were found to be equally distributed in Santa Catarina between May 2021 and April 2022. This is in contrast with the findings of Padilha et al. (2022), who also compared different regions of Santa Catarina and showed a relationship between the variant (P.1-like-II) and region (Western), associating this factor with increased propagation of the disease and a higher mortality rate [15].

In Brazil, Scheler et al. (2022) compared the deaths of pregnant and non-pregnant women during the first and second waves of the pandemic [24]. The authors demonstrated that maternal deaths from COVID-19 increased in the second wave compared with the first wave, and this increase was greater in patients with comorbidities. Another study in Brazil compared mortality among different age groups between January 2021 and February 2022. During this study, approximately 400 million deaths were analyzed, and the same findings showed that the second wave had a higher mortality rate than the first wave [25]. de Souza et al. (2021) demonstrated the high prevalence of the Gamma variant in the state of Amazonas (Northern Brazil) [26]. It is possible to correlate the presence of this variant with an increase in the number of deaths. Several hypotheses exist for these findings, such as the difficulty in enforcing social distancing, incorrect application of nonpharmaceutical interventions, and low or late vaccination, particularly in young populations. The present study corroborates the findings of Scheler and Orellana, considering that during the second pandemic wave, it was also possible to observe an increase in the number of deaths (Figure 1B), with peaks in May 2021 and January 2022, both related to the appearance of Gamma and Omicron variants (Figure 1C,D), respectively. Unlike the Gamma variant, which presented high mortality even with a low number of cases, the Omicron variant was characterized by a greater number of cases than the number of deaths in this study.

Omicron variants are characterized by approximately 30 mutations in the Spike (S) protein alone, while other mutations are spread throughout the rest of the viral genome. These mutations appear to increase the affinity of the S protein for the host angiotensin-converting enzyme 2 (ACE2) receptor, which may be related to the higher transmissibility of Omicron compared with other SARS-CoV-2 variants [14,27]. This hypothesis would explain the findings of this study, in which we observed an increase in the number of cases, but not in the number of deaths, during the period in which the Omicron variant was more prevalent (Figure 1A,B). The holidays’ celebrations, the flexibility of nonpharmaceutical interventions (NPI) measures and the summer vacation period in Santa Catarina State at the end of 2021 could also contribute to the crowded gatherings in closed spaces and the increase in the number of cases in this period of 2021, as observed previously in other countries [28,29,30] and in other Brazilian states [31,32].

In Italy, during the first wave of COVID-19, HCW were more easily infected with SARS-CoV-2 than the general population, and their infection was independent of whether they were operating in COVID-19 or non-COVID-19 working environments [3]. As soon as vaccination against SARS-CoV-2 became available, governments prioritized the vaccination of HCW. In Southern Brazil, including Santa Catarina, vaccination began in early 2021, and HCW were vaccinated so they may perform their jobs on-site. There is no consensus on adherence to vaccination in the general population in most regions of the country. Differences in vaccination status may have influenced the delay in the initial detection of SARS-CoV-2 in HCW compared with the general population (Figure 2). As a result, fewer samples were collected from HCW until epiweek 52 compared to the general population, and consequently, fewer varieties in strain lineages (only Delta lineages) were seen. Guijarro et al. (2021) observed that after the first dose of BNT162b2 mRNA SARS-CoV-2 vaccination of HCW, the SARS-CoV-2 incidence decreased by 71% (symptomatic and asymptomatic tested) compared to the 2% in the non-vaccinated general population [33]. The HCW sample collection in our research began in March 2021, after the start of their vaccination. In the following weeks of monitoring, there were no differences between the lineages detected in either group (Figure 3 and Appendix A), with Omicron lineages becoming dominant in the epidemiological scenario. This demonstrates that despite the medical professionals being exposed more during patient care, they may be better protected owing to the use of personal protective equipment and the efficacy of vaccine adherence.

The total of HCW, the restricted study period and the regionality are limiting factors of the study. Therefore, to generalize the findings of this work, with an error rate of 9.3 at 95% confidence level, these limit factors must be considered.

## 5. Conclusions

Based on the data presented here, it is possible to reinforce the importance of genomic surveillance as a tool for monitoring SARS-CoV-2 infection. Our findings support previous literature demonstrating a surge in COVID-19 cases during the second wave of the pandemic, and that the two notable peaks in deaths were associated with the emergence of the Gamma and Omicron VOCs. However, we should point out that the change from Gamma to Delta variant was slower in Brazil than in other countries, and the daily cases decreased. In addition, we showed that the VOC Omicron became the most predominant VOC in Santa Catarina four weeks after the first case, and it was correlated with the highest number of cases but not the highest number of deaths. Moreover, our data revealed that HCW are a critical sentinel group for the genomic surveillance of SARS-CoV-2, given that the VOC profile is similar to that observed in the general population and the previous VOC shifts in this group. Taken together, our results demonstrate that HCW are critical representatives of the major changes in variant profiles in the general population. This is due to the similar behavior of these two groups in terms of viral genome dispersion. Few studies have investigated HCW as a sentinel group for determining the current trends in emerging SARS-CoV-2 variants and their pathogenesis. Our study is a pioneering one in Brazil for considering this approach for the genomic surveillance of SARS-CoV-2. Identifying representative sample groups of the population may bring greater accuracy to data in regions with limited investments, making this type of surveillance more feasible.

## Figures and Tables

**Figure 1 viruses-15-00984-f001:**
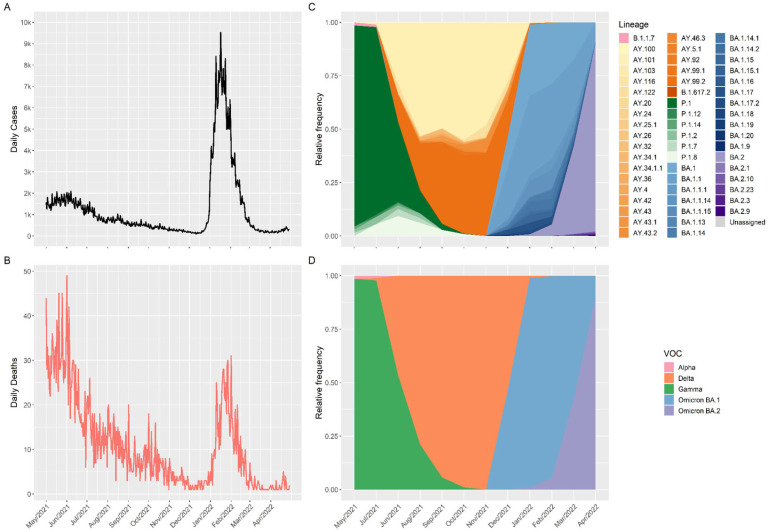
Number of cases/deaths and variant distribution of SARS-CoV-2 from 1 May 2021 to 30 April 2022. (**A**) Daily cases and (**B**) daily deaths, according to the Boa Vista platform (https://dados.sc.gov.br, accessed on 15 December 2022). SARS-CoV-2 variant profile (**C**) by lineage and (**D**) by variant of concern (VOC).

**Figure 2 viruses-15-00984-f002:**
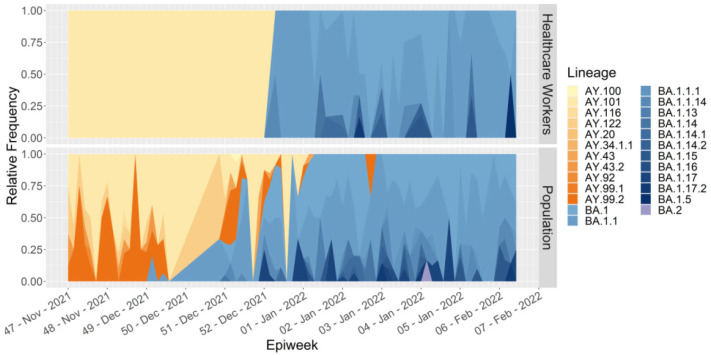
Comparison of relative frequency for each SARS-CoV-2 lineage in healthcare workers and general population during the variant shift from Delta to Omicron.

**Figure 3 viruses-15-00984-f003:**
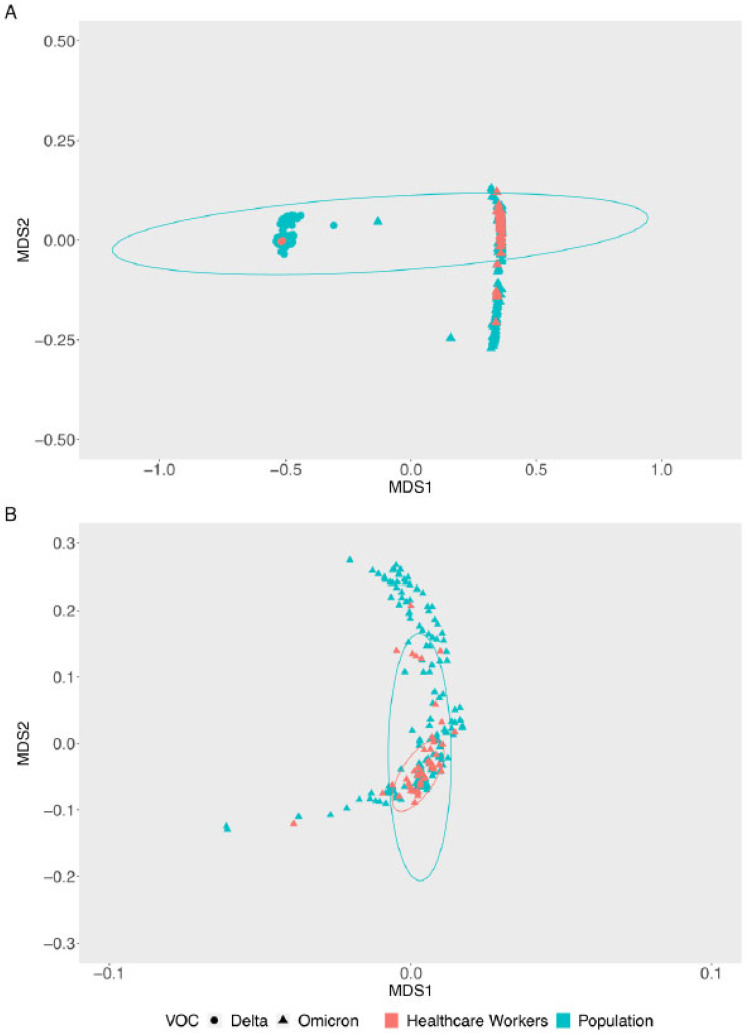
Multidimensional scale (MDS) with Jaccard dissimilarity plot of genomes based on amino acids and presence/absence of mutations, deletions, and insertions—a comparison between healthcare workers and the general population of Santa Catarina, from November 2021 to February 2022. (**A**) Displaying Delta and Omicron variants (**B**) Omicron BA.1 lineage and sublineages.

## Data Availability

All DNA sequences generated during the current study are available in the GISAID EpiCoV repository (https://www.gisaid.org, accessed on 14 January 2023) under the following EPI_SET IDs: *EPI_SET_230321ms* (samples sequenced by our research group) and *EPI_SET_230321kb* (samples originated by other research groups).

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
