# Peer review of "Genomic Surveillance of SARS-CoV-2 in Healthcare Workers: A Critical Sentinel Group for Monitoring the SARS-CoV-2 Variant Shift"

_viruses, 2023, doi:10.3390/v15040984_

Round 1

Reviewer 1 Report

This study shows that SARS-CoV-2 genome surveillance is important for monitoring risk groups and health workers. The Authors aimed to evaluate the dispersion of SARS-CoV-2 among the general population in Santa Catarina, Brazil, from May 2021 to April 2022 and compare it to that in HCW during the Omicron wave to understand the role of HCW as sentinels in human viral surveillance.

This study shows  interesting data, is well structured and well written. The methods are clear and well detailed. The conclusions are adequately supported by the results and interpretation of the data. The references are appropriate and current. The figures (including the supplemental figure) are interesting, very specific, not of immediate interpretation, but satisfactory.

The only suggestion I make to the Authors is to include the limitations of this study.

Author Response

Dear Reviewer,

We appreciated your comments.

Comment I: “The only suggestion I make to the Authors is to include the limitations of this study.”

We are flattered by the mentioned comments and agree with the importance of including the limitations of this study. With that, we added a paragraph, on page 9, at the end of the discussion, which mentions such limitations as follows:

The total of HCW, the restricted study period and the regionality are limiting factors of the study. Therefore, to generalize the findings of this work, with an error rate of 9.3 at 95% confidence level, these limit factors must be considered.”.

Please see the revised article attached. 

best regards.

Glauber Wagner

Reviewer 2 Report

Dear Authors,

The manuscript entitled "Genomic surveillance of SARS-CoV-2 in healthcare workers: a critical sentinel group for monitoring the SARS-CoV-2 variant shift" has been reviewed.

This article deserves attention since it highlights a very important topic related to the monitoring of SARS-CoV-2 variants shift in Healthcare workers and the population in Santa Catarina-Brasil, using a genomic surveillance of this virus. In fact this important study aims to a find representative samples that can help public health officers in reduce the costs associated with genomic surveillance.

The present article is well written in English language and well presented, but I have some comments (minors and majors) regarding it:

Minor Comments:

01- In the whole manuscript, mainly for the next submission, authors are invited to add Line numbers.

02- In the whole manuscript, authors are invited to put the term "et al.," in italic.

03- In the Keywords section, authors are invited to replace the term "variant" by "SARS-CoV-2 variants".

04- In the Introduction section, at the end of this section, authors have to talk in general about the fact that different studies show that the variants have a higher transmission rate or infective rate than the wild type. They are invited to add the following reference: The emergence of SARS-CoV-2 variant (s) and its impact on the prevalence of COVID-19 cases in the Nabatieh Region, Lebanon.

05- In the Materials and Methods section, some words are missing in the following sentence "An error rate of 9.31% was obtained at a 95% confidence level. The sample size was 110 and the total number of HCW (N) was 15,382, assuming a homogeneous distribution, with" !!

Major comments:

01- In the Materials and Methods section, authors mentioned that they received the approval from the Ethics Committee on October 29th, 2021. then in the next paragraph they talk about the collection of samples between May 2021 and April 2022, so they are invited to explain how do they start the study (in May 2021) before that the study was approved by the Ethics Committee (in October 2021)?

02- In the Discussion section, Authors are invited to talk about the fact that the increase of positive cases number during December and January in the study period may be related to Christmas’ and holidays’ celebrations due to crowded gatherings in closed spaces. They are invited to add this idea with the following reference: Risk Markers of COVID-19, a Study from South-Lebanon.

Best Regards,

Author Response

Dear Reviewer,

We appreciate your essential comments. Please see below ours responses of your coments.   Minor Comments:

Comment 01 - In the whole manuscript, mainly for the next submission, authors are invited to add Line numbers.

Answer: We appreciated the observation. The lines were not added because we used the journal template in which the lines number are not included. However, we will include in the next submissions.

Comment 02- In the whole manuscript, authors are invited to put the term "et al.," in italic.

Answer: We appreciated your observation and changed the term in the revised manuscript accordingly.

 Comment 03- In the Keywords section, authors are invited to replace the term "variant" by "SARS-CoV-2 variants".

Answer: We appreciate your observation. We replaced the keyword to “SARS-CoV-2 variants” in the revised manuscript.

Comment 04- In the Introduction section, at the end of this section, authors have to talk in general about the fact that different studies show that the variants have a higher transmission rate or infective rate than the wild type. They are invited to add the following reference: The emergence of SARS-CoV-2 variant (s) and its impact on the prevalence of COVID-19 cases in the Nabatieh Region, Lebanon.

Answer: We appreciate your comment. To enrich the discussion of the present study, we preferer to include the recommended references in the discussion on page 8.

The holiday's celebrations, the flexibility of nonpharmaceutical interventions (NPI) measures and the summer vacation period in Santa Catarina State at the end of 2021 could also contribute to the crowded gatherings in closed spaces and the increase in the number of cases in this period of 2021, as observed previously in other countries [29,30,31] and in other brazilian States [32,33].”

Comment 05 - In the Materials and Methods section, some words are missing in the following sentence "An error rate of 9.31% was obtained at a 95% confidence level. The sample size was 110 and the total number of HCW (N) was 15,382, assuming a homogeneous distribution, with" !!

Answer: We appreciate your observation. We corrected it in the revised manuscript.

Major comments:

Comment 01- In the Materials and Methods section, authors mentioned that they received the approval from the Ethics Committee on October 29th, 2021. then in the next paragraph they talk about the collection of samples between May 2021 and April 2022, so they are invited to explain how do they start the study (in May 2021) before that the study was approved by the Ethics Committee (in October 2021)?

Answer: We appreciate your comment. In this study, we sequenced samples from the General Population and HCW between October 2021 and April 2022 after approval by the UFSC ethics committee. To complement the study, we included public data deposited on the GISAID platform since May 2021. Thus, the data analyzed between May and September 2021 were exclusively obtained from the data available on GISAID. To make it clearer in the text, we added the following sentence in the section 2.1 Studied Regions and Sampling”: “Only public data available between May 2021 and September 2021 on the GISAID platform were analyzed.”

Comment 02- In the Discussion section, Authors are invited to talk about the fact that the increase of positive cases number during December and January in the study period may be related to Christmas’ and holidays’ celebrations due to crowded gatherings in closed spaces. They are invited to add this idea with the following reference: Risk Markers of COVID-19, a Study from South-Lebanon.

Answer: We included the discussion and the references recommended in the following paragraph on page 8. We discuss that besides the Omicrom variant characteristic, the increase in the number of SARS-CoV-2 cases between December 2021 to April 2022 could be a consequence of the flexibility of lockdown measures and the holiday’s events by the end of 2021.

The holidays' celebrations, the flexibility of nonpharmaceutical interventions (NPI) measures and the summer vacation period in Santa Catarina State at the end of 2021 could also contribute to the crowded gatherings in closed spaces and the increase in the number of cases in this period of 2021, as observed previously in other countries [29,30,31] and in other Brazilian States [32,33].”

Please find attached the revised manuscript.

Best regards,

Glauber Wagner

Round 2

Reviewer 2 Report

Dear Authors,

Your manuscript has been re-reviewed.

The new version of the article is better.

Thank you for the modifications you did.

Best Regards